# Selective hydrosilylation of allyl chloride with trichlorosilane

Koya Inomata[1], Yuki Naganawa [1], Zhi An Wang[1,2], Kei Sakamoto[1], Kazuhiro Matsumoto [1], Kazuhiko Sato[1] & Yumiko Nakajima [1,2 ✉]

The transition-metal-catalysed hydrosilylation reaction of alkenes is one of the most important catalytic reactions in the silicon industry. In this field, intensive studies have been thus far performed in the development of base-metal catalysts due to increased emphasis on environmental sustainability. However, one big drawback remains to be overcome in this field: the limited functional group compatibility of the currently available Pt hydrosilylation catalysts in the silicon industry. This is a serious issue in the production of trichloro(3-chloropropyl)silane, which is industrially synthesized on the order of several thousand tons per year as a key intermediate to access various silane coupling agents. In the present study, an efficient hydrosilylation reaction of allyl chloride with trichlorosilane is achieved using the Rh(I) catalyst $[RhCl(dppbz^F)]_2$ ($dppbz^F$ = 1,2-bis(diphenylphosphino)-3,4,5,6-tetra-fluorobenzene) to selectively form trichloro(3-chloropropyl)silane. The catalyst enables drastically improved efficiency (turnover number, TON, 140,000) and selectivity (>99%) to be achieved compared to conventional Pt catalysts.

[1] Interdisciplinary Research Center for Catalytic Chemistry, National Institute of Advanced Industrial Science and Technology (AIST), Tsukuba, Ibaraki, Japan. [2] Graduate School of Pure and Applied Sciences, University of Tsukuba, Tsukuba, Ibaraki, Japan. ✉email: yumiko-nakajima@aist.go.jp

The hydrosilylation reaction, which achieves the addition of hydrosilanes to alkenes, is one of the most important catalytic reactions in the silicon industry[1–6]. For more than half a century, this reaction has been employed in the production of various organosilicon compounds that are used as synthetic precursors as well as in the curing of silicone products[1–3,7,8]. Another important application of the hydrosilylation reaction is the production of silane coupling agents, which has recently elevated the importance of this reaction greatly. Silane coupling agents, which are usually γ-functionalised propyl silanes of type X (CH$_2$)$_3$Si(OR)$_3$ (X = various functional groups), have the ability to form a durable bond between organic and inorganic materials and to provide the resulting compounds or materials with various properties, such as water and/or heat resistance as well as adhesiveness, without negatively affecting the original properties of either the organic or the inorganic materials[9–12]. By taking advantage of such features, silane coupling agents are currently used in a great number of fields including paints, coating, adhesives, semiconductor sealants, and tires.

Trichloro(3-chloropropyl)silane (1) is an important key intermediate to access various silane coupling agents via simple nucleophilic substitution reaction and alcoholysis of trichlorosilyl group[13]. Thus, 1 is now industrially synthesised on the order of several thousand tons per year via a simple hydrosilylation reaction, i.e. the reaction of allyl chloride with HSiCl$_3$ catalysed by conventional Pt catalysts, such as Speier's catalyst[14] or Karstedt's catalyst[15] (Fig. 1a). However, the relatively low selectivity of the reaction represents a severe drawback[6,16–24]. Pt-catalysed hydrosilylation reactions usually proceed through a (modified) Chalk–Harrod mechanism[25,26], which is initiated with the oxidative addition of a hydrosilane to a Pt metal centre, followed by alkene insertion and reductive elimination (Fig. 1b, path 1). In contrast, allyl chloride also exhibits a high propensity to engage in the oxidative addition reaction to form stable π-allyl Pt species; thus, the oxidative addition of two substrates, hydrosilane and allyl chloride, proceeds competitively during the hydrosilylation reaction. Consequently, several side products, including trichloropropylsilane (2), propene and SiCl$_4$, are formed during the hydrosilylation of allyl chloride (Fig. 1b, path 2)[6,16–24], which causes difficulties in the purification process and sometimes, degradation in the performance of the materials. Given the increasing demand for the precise synthesis of various organosilicon compounds to achieve more sophisticated high-performance materials in recent years, the development of novel catalytic systems that can efficiently catalyse the hydrosilylation of allyl chloride and other allylic compounds with various functionalities is highly desirable[6,27–35]. To date, several mechanistic studies on the hydrosilylation of allyl chloride have been reported[18–20]; however, this industrially important issue has not been sufficiently discussed in academic research, despite the recent dramatic progress in the development of base-metal catalysts for the hydrosilylation of simple alkenes[36–39].

In this study, we achieved efficient hydrosilylation of allyl chloride with HSiCl$_3$ using Rh catalysts that bear bidentate-phosphine ligands[40]. The precise design of these catalysts, which was developed based on a detailed mechanistic study, successfully improved the performance of the catalysis and enabled excellent efficiency and selectivity (a TON of 140,000 at 5 ppm/Rh and >99% selectivity) to be achieved.

## Results

**Screening of metal catalysts.** The optimisation of the reaction of allyl chloride with HSiCl$_3$ was performed at a catalyst loading of 0.5 mol%/metal at 60 °C for 3 h (Table 1 and Supplementary Fig. 1). Our preliminary experiments using conventional Pt catalysts clearly showed their poor product selectivity. The use of Speier's and Karstedt's catalysts resulted in the formation of the hydrosilylated product 1 in low yields of 20% and 15%, respectively (Table 1, entries 1 and 2). The reactions were accompanied by the formation of by-product 2 in 32% and 13% yields, respectively. Additional formation of propane was also confirmed based on the $^1$H NMR spectra of the reaction mixture. Both the reaction efficiency and selectivity were somewhat improved using a combination of Karstedt's catalyst and 1,3-dimesitylimidazol-2-ylidene (IMes) (2 equiv/Pt)[41], which afforded 1 (53%) and 2 (14%) (Table 1, entry 3). The screening of various metal precursors was performed (Supplementary Table 1). Surprisingly, Ir catalysts, which have been reported to be useful catalysts for hydrosilylation of allyl chloride with HSi(OR)$_3$ or HSiClMe$_2$ hardly catalysed the reaction[24,34,42–44]. On the other hand, the reaction in the presence of classical Wilkinson's catalyst [RhCl(PPh$_3$)$_3$][45–48] proceeded with better reaction selectivity to give 1 in 26% yield and 2 in <5% yield (Table 1, entry 4). Therefore, we then moved on to the investigation of Rh catalysts prepared in situ via the reaction of [Rh($\mu$-Cl)(cod)]$_2$ and various phosphine ligands. For example, similar results to those obtained with Wilkinson's catalyst were achieved using a combination of [Rh($\mu$-

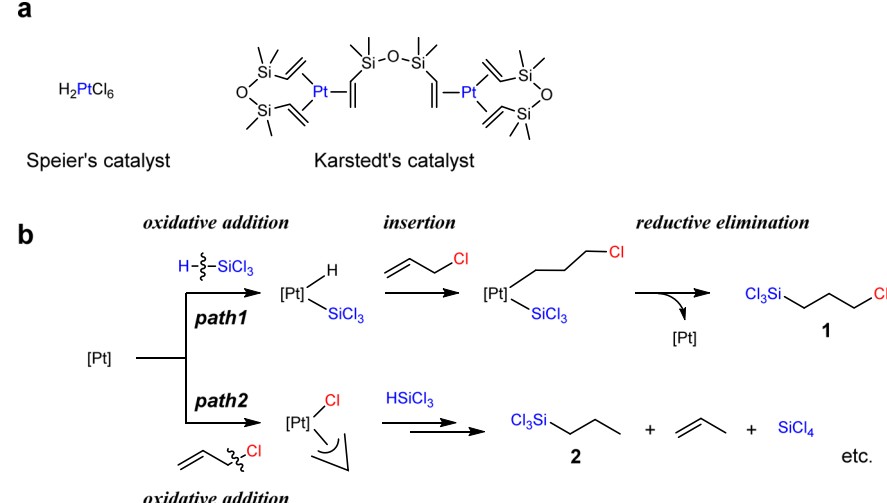

**Fig. 1 Pt-catalysed hydrosilylation reaction. a** Conventional Pt hydrosilylation catalysts. **b** Formation pathways for the desired product **1** (path 1) and by-products (path 2).

**Table 1 Screening of metal catalysts in the hydrosilylation of allyl chloride with $HSiCl_3$.**

| Entry | Catalyst | Ligand (x) | %Yield (1)[a] | %Yield (2)[a] |
|---|---|---|---|---|
| 1 | Speier's catalyst[b] | None | 20 | 32 |
| 2 | Karstedt's catalyst | None | 15 | 13 |
| 3 | Karstedt's catalyst | IMes (1) | 53 | 14 |
| 4 | Wilkinson's catalyst | None | 26 | <5 |
| 5 | [Rh(μ-Cl)(cod)]₂ | PPh₃ (1) | 31 | <5 |
| 6 | [Rh(μ-Cl)(cod)]₂ | PCy₃ (1) | 45 | <5 |
| 7 | [Rh(μ-Cl)(cod)]₂ | dppe (0.5) | 76 | <5 |
| 8 | [Rh(μ-Cl)(cod)]₂ | dppp (0.5) | 88 | 6 |
| 9 | [Rh(μ-Cl)(cod)]₂ | dppbz (0.5) | 93 | 7 |

Reaction conditions: Allyl chloride (1.0 mmol), HSiCl₃ (1.0 mmol), and catalyst (0.005 mmol/metal) under a nitrogen atmosphere.
[a]Determined by ¹H NMR spectroscopy using mesitylene as an internal standard.
[b]H₂PtCl₆.

**Fig. 2 Stoichiometric reaction of [RhCl(cod)]₂ with dppp or dppbz and catalytic activities of the resulting complexes 3–7. a** Formation of **3** and **4**. **b** Formation of **5**, **6**, and **7**. **c** Hydrosilylation reaction of allyl chloride catalysed by **3**–**7**.

Cl)(cod)]₂ and PPh₃ (2 equiv/Rh) to furnish **1** and **2** in 31% and <5% yields, respectively (Table 1, entry 5). The reaction catalysed by [Rh($\mu$-Cl)(cod)]₂ and electron-donating PCy₃ ligand gave **1** with a slightly improved yield of 45% (Table 1, entry 6). Further studies revealed that the use of bidentate-phosphine ligands 1,2-bis(diphenylphosphino)ethane (dppe) and 1,3-bis(diphenylphosphino)propane (dppp) significantly improved the reaction yield and selectivity, forming **1** in 76% and 88% yields, respectively (Table 1, entries 7 and 8). The highest yield of **1** (93%) was achieved with 1,2-bis(diphenylphosphino)benzene (dppbz), which was accompanied by the formation of **2** in 7% yield (Table 1, entry 9). The formation of propene was not observed by ¹H NMR spectroscopy in any of these Rh-catalysed reactions.

**Identification of the catalytically active rhodium complexes.** As shown in Table 1, entries 7–9, the choice of the Rh precursor and bidentate ligand combination had a significant influence on the catalytic performance in the systems. To shed light on such diverse effects of the bidentate ligand, stoichiometric reactions of [Rh($\mu$-Cl)(cod)]₂ with the promising ligands, dppp and dppbz, were performed. When the reaction of [Rh($\mu$-Cl)(cod)]₂ with dppp (1 equiv/Rh) was performed at 60 °C for 3 h, the formation of a mixture composed of two complexes, [Rh($\mu$-Cl)(dppp)]₂ (**3**) (79%) and [RhCl(cod)(dppp)] (**4**) (6%), was confirmed by ¹H and ³¹P{¹H} NMR spectroscopic analysis (Fig. 2a). In contrast, the reaction of [Rh($\mu$-Cl)(cod)]₂ with dppbz (1 equiv/Rh) resulted in the formation of [Rh($\mu$-Cl)(dppbz)]₂ (**5**) (63%), [(dppbz)Rh($\mu$-Cl)₂Rh(cod)] (**6**) (26%), and [Rh(dppbz)₂]Cl (**7**) (8%; Fig. 2b). According to these results, we realised that mixing the Rh precursor [Rh($\mu$-Cl)(cod)]₂ and a ligand resulted in the in situ formation of various Rh species with a different molar ratio, which are dependent on the used bidentate ligand. Thus, we examined the catalytic performance of each complexes (Fig. 2c and Supplementary Table 2) which have been independently synthesised according to the alternative procedures (see Supplementary Methods section, page 15 of the Supplementary Information). Complexes **3** and **4** catalysed the hydrosilylation of allyl chloride with HSiCl₃ at the catalyst loading of 0.5 mol%/Rh, affording **1** in 60% and 41% yields, respectively. The reaction catalysed by **5** proved to be most efficient among **3**–**7** under otherwise identical

reaction conditions to form **1** in >95% yield. Complex **6**, in which only one of the two Rh centres was supported by dppbz, exhibited the significantly suppressed catalytic activity compared to **5**. Likewise, complex **7** coordinated with two dppbz ligands hardly catalysed the reaction. In all the above reactions, the formation of only trace amounts of **2** was confirmed. Based on these results, we postulated that chloro-bridged dinuclear complex **5**, which could form a reactive mononuclear dppbz-Rh (I) species 'RhᴵCl (dppbz)', behave as a good catalyst precursor.

**Investigation of effective bidentate-phosphine ligands.** Based on the observations discussed above, it became clear that a rigorous evaluation of the ligand effect on the catalytic efficiency should be carried out using independently prepared Rh complexes coordinated with a ligand rather than catalysts prepared in situ by mixing a Rh precursor and a ligand. Accordingly, we then surveyed the catalytic activity of a series of chloro-bridged Rh dimers that bear bidentate-phosphine ligands[49,50]; the results are summarised in Table 2 (Supplementary Table 3). All the examined reactions resulted in the selective formation of **1** accompanied by only trace amounts of by-product **2**. The reactions catalysed by 500 ppm/Rh of [Rh($\mu$-Cl)(dppe)]₂, [Rh($\mu$-Cl)(dppp)]₂ (**3**), or [Rh($\mu$-Cl)(dppb)]₂ (dppb = 1,4-bis(diphenylphosphino)butane), which possess ligands with an alkyl backbone, were low yielding, affording **1** in 3%, 13%, and 22% yields, respectively, at 60 °C after 20 h (Table 2, entries 1–3). As demonstrated in the experiments shown in Fig. 2c, the use of [Rh($\mu$-Cl)(dppbz)]₂ (**5**) delivered a better performance (Table 2, entry 4). Using **5**, the quantitative formation of **1** was achieved at a lower catalyst loading of 50 ppm/Rh, accompanied by the formation of a trace amount of **2**. At a further decreased catalyst loading of **6** (5 ppm/Rh), the reaction afforded **1** in 11% yield (Table 2, entry 5).

We then examined the substituent effect with respect to the diphenylphosphino moieties using catalysts **8** and **9**; thus, the introduction of either electron-donating or electron-withdrawing substituents resulted in diminished catalyst performance, leading to the formation of <5% yield of **1** (Table 2, entries 6 and 7). Next, the substituent effect with respect to the phenylene backbone was examined. The use of catalyst [Rh($\mu$-Cl)(dppbzᴼᴹᵉ)]₂ (**10**), which

**Table 2 Hydrosilylation of allyl chloride with HSiCl₃ catalysed by Rh catalysts bearing a bidentated phosphine ligand.**

| Entry | Cat (ppm/Rh) | Temp. (°C) | Time (h) | %Yield (1)[a] |
|---|---|---|---|---|
| 1 | [Rh(μ-Cl)(dppe)]₂ (500) | 60 | 20 | 3 |
| 2 | [Rh(μ-Cl)(dppp)]₂ (**3**) (500) | 60 | 20 | 13 |
| 3 | [Rh(μ-Cl)(dppb)]₂ (500) | 60 | 20 | 22 |
| 4 | [Rh(μ-Cl)(dppbz)]₂ (**5**) (50) | 60 | 20 | >95 |
| 5 | **5** (5) | 60 | 20 | 11 |
| 6 | **8** (50) | 60 | 20 | <5 |
| 7 | **9** (50) | 60 | 20 | <5 |
| 8 | **10** (50) | 60 | 20 | <5 |
| 9 | **11** (50) | 60 | 20 | >95 |
| 10 | **11** (50) | 25 | 10 | 73 |
| 11 | **11** (50) | 40 | 20 | 29 |
| 12 | **11** (50) | 60 | 20 | 39 |
| 13 | **11** (5) | 60 | 20 | 29 |
| 14[b] | **11** (5) | 60 | 20 | 70 |

Reaction conditions: Allyl chloride (1.0 mmol), HSiCl₃ (1.0 mmol) and the Rh catalyst under a nitrogen atmosphere.
[a]Determined by ¹H NMR spectroscopy using mesitylene as an internal standard.
[b]Using 3 equiv of HSiCl₃.

**Fig. 3 Reactivity study on π-allyl complexes bearing a bidentate-phosphine ligand. a** Reaction of **11** with allyl chloride. **b** Reaction of **12** with HSiCl₃. **c** Formation pathways for the by-products SiCl₄ and **2** (path 1) and the desired product **1** (path 1). **d** Reaction of **12** with cinnamyl chloride. **e** Reaction of **13** with HSiCl₃.

carries 1,2-bis(diphenylphosphino)-3,4-dimethoxybenzene ligands with electron-donating methoxy groups on the phenylene backbone, resulted in the only slight formation of **1** (Table 2, entry 8). In contrast, the use of complex [Rh(μ-Cl)(dppbz$^F$)]₂ (**11**), bearing 1,2-bis(diphenylphosphino)-3,4,5,6-tetrafluorobenzene (dppbz$^F$) with a perfluorophenylene backbone, achieved improved catalytic activity. Thus, the hydrosilylation reaction of allyl chloride with HSiCl₃ proceeded in the presence of 50 ppm/Rh of **11** to form **1** in >95% yield at 60 °C after 20 h (Table 2, entry 9). The good yield of 73% (**1**) was also achieved after the diminished reaction time, 10 h (Table 2, entry 10). Whereas, significant decrease in the yields of **1** was confirmed at a lower temperature (Table 2, entries 11, 12). Gratifyingly, **11** exhibited higher catalytic activity than **5**; i.e. the reaction with 5 ppm/Rh of **11** resulted in the formation of 29% yield of **1** (Table 2, entry 13). The yield of **1** further increased to 70%, and only a trace amount

of **2** was formed as a side product when 3 equiv of HSiCl₃ was used (Table 2, entry 14) (it was confirmed that the yield of **1** was not improved after further reaction time). Under these reaction conditions, the TON of 140,000 was achieved.

**Mechanistic considerations.** In the Rh-catalysed reactions (Tables 1 and 2), the concomitant formation of **2** was constantly observed albeit in low yields. These results strongly suggest the occurrence of oxidative addition of allyl chloride to the Rh centre during the catalytic reactions[51–53]. To further clarify this point, the reactivity of **11** towards allyl chloride was investigated. Upon treatment with allyl chloride (1 equiv/Rh), **11** was converted into the corresponding oxidative addition product, [Rh(π-allyl)Cl₂(dppbz$^F$)] (**12**), at ambient temperature (Fig. 3a). The resulting complex **12** further reacted with excess HSiCl₃ at 60 °C to form **1** (26%) and **2** (45%), together with other unidentified

**Fig. 4 Plausible catalytic cycle for the Rh-catalysed hydrosilylation of allyl chloride with HSiCl₃.** Reaction path for a selective formation of the hydrosilylated product.

complexes (Fig. 3b, see Supplementary Information, Supplementary Methods, page 13). The formation of $SiCl_4$ was also confirmed by $^{29}Si\{^1H\}$ NMR spectroscopic analysis. This result clearly demonstrates the path by which **2** was formed during the catalysis (Fig. 3c, path 1): **12** reacts with $HSiCl_3$ to form $[RhCl_2(SiCl_3)(propene)(dpppbz^F)]$[54,55], which undergoes reductive elimination to form the catalytically active 'Rh$^I$Cl(dppbz$^F$)' species and $SiCl_4$. The subsequent hydrosilylation of propene with $HSiCl_3$ results in the formation of **2**. Another remarkable feature of this reaction is the formation of **1** (26%) in addition to **2**. This observation strongly suggests the occurrence of reductive elimination of allyl chloride from **12** to reproduce 'Rh$^I$Cl(dppbz$^F$)' species, which would successively facilitate hydrosilylation of allyl chloride to form **1** (Fig. 3c, path 2). Such the fluctuating ligand behaviour in **12** was also supported by the reaction of **12** with cinnamyl chloride. In the presence of 20 equiv of cinnamyl chloride, **12** was partially transformed to an unidentified complex **A** (ca. 23% conversion), which could be assignable to [Rh(π-cinnamyl)Cl₂(dppbz$^F$)] based on ESI-Mass spectroscopic analysis. In addition, the formation of free allyl chloride (20% NMR yield) was confirmed (Fig. 3d, see Supplementary Information, Supplementary Methods, page 13). The reaction is likely to proceed via the ligand exchange between the allyl ligand and the cinnamyl ligand, thus demonstrating reversible oxidative addition and reductive elimination pathway of allyl chloride.

In contrast, the reaction of [Rh(π-allyl)Cl₂(dppp)] (**13**) with $HSiCl_3$ resulted in the quantitative formation of **2**, accompanied by the formation of trace amounts of propene and $SiCl_4$ (Fig. 3e). The reaction also afforded [Rh(Cl)(H)(SiCl₃)(dppp)] (**14**) (91% NMR yield), which should be formed via the oxidative addition of $HSiCl_3$ to the 'Rh$^I$(dppp)' species that is generated in situ. Indeed, **14** could alternatively be synthesised by the reaction of [Rh(μ-Cl)(dppp)]₂ (**3**) with $HSiCl_3$. Overall, these experimental data strongly support the conclusion that the dppbz$^F$ ligand facilitates the reductive elimination of allyl chloride, possibly due to its effective stabilisation of the electron-rich Rh(I) centre.

**Proposed catalytic cycle.** Based on the experimental results described above as well as on the previous reports[23,24,47,56], we postulate a plausible reaction pathway that proceeds in the presence of **11** following a modified Chalk–Harrod mechanism, which is the typical mechanism for Rh-catalysed hydrosilylation reactions (Fig. 4)[56–62]. In this pathway, the dinuclear Rh(I) precursor **11** reacts with $HSiCl_3$ to form hydrido(silyl)Rh(III) complex **B**, which successively undergoes insertion of allyl chloride

into the Rh–Si bond. The reductive elimination of the Si–H bond regenerates **11**, furnishing the hydrosilylated product. Complex **11** also engage in the oxidative addition of allyl chloride to form **12**. The feasibility of this step was confirmed by the mechanistic study discussed above (Fig. 3a). Our mechanistic study suggested that dppbz$^F$ facilitates the reductive elimination of allyl chloride from **12** because of its stabilisation of the Rh(I) centre, which in turn enables the selective formation of **1**. It could also be possible that electron-withdrawing dppbz$^F$ ligand facilitates the insertions step[63,64], which is known to be the rate-determining step in the modified Chalk–Harrod process[55–62]. To further shed light on the effect of dppbz$^F$ ligand, a detailed mechanistic study based on DFT study is now underway in our laboratory. Considering the structure of **14**, which possesses the hydrido ligand on the basal plane, there is a possibility that the reaction could follow Chalk–Harrod mechanism, where allyl chloride inserts into the Rh–H bond. This point is also a target of our interest in the DFT study.

**Gram-scale synthesis.** With this powerful catalyst in hand, we attempted the gram-scale synthesis of **1**. The reaction of 2.5 mol of allyl chloride with 1 equiv of $HSiCl_3$ in the presence of 50 ppm/Rh of **11** afforded 483 g (2.28 mol) of **1** with excellent selectivity (>99%), which was easily isolated as a pure form after simple distillation procedure (Fig. 5).

This study established the superior utility phosphine-supported Rh catalysts to the conventional Pt catalysts in an industrially important hydrosilylation reaction, i.e. the selective hydrosilylation of allyl chloride with $HSiCl_3$ to form trichloro(3-chloropropyl)silane (**1**). The precise design of the catalyst structure enabled the development of the new catalyst [Rh(μ-Cl)(dppbz$^F$)]₂ (**11**), which achieved excellent activity and selectivity in the reaction. A detailed mechanistic study revealed that the major side product trichloropropylsilane (**2**) is formed through successive reactions of the catalyst with allyl chloride and $HSiCl_3$, followed by hydrosilylation of the propene that is generated in situ. This mechanistic study also supports the notion that dppbz$^F$, with an electron-withdrawing backbone, is able to stabilise the electron-rich Rh(I) species and thereby suppress the formation of **2**. This fundamental study has led to a dramatic improvement in the selectivity of the reaction and enables the industrially important key compound **1** to be furnished efficiently. This achievement represents a significant step towards resolving one of the biggest unsolved problems in industrial hydrosilylation chemistry, i.e. the limited functional group compatibility of conventionally employed Pt catalysts. The application of the method developed in this study allows the synthesis of 475 g of **1** in pure form. We are convinced that the results of this work will pave the way to new avenues and inspire further research in the area of practical hydrosilylation chemistry, which would translate to innovations in the silicon industry.

## Methods

**[Rh(μ-Cl)(dppbz$^F$)]₂ (11) catalysed hydrosilylation reaction of allyl chloride with HSiCl₃.** Toluene stock solution of **11** (10 μL, 0.0025 μmol, 0.5 mM) was added to a 10 mL screw vial equipped with a stir bar under a nitrogen atmosphere. After removal of toluene in vacuo, allyl chloride (81 μL, 1.0 mmol), and trichlorosilane (300 μL, 3.0 mmol) were added to the vial. The mixture was stirred at 60 °C for 20 h. After the reaction, mesitylene (14 μL, 0.10 mmol) was added to the mixture as an internal standard and the yield of 3-chloropropyltrichlorosilane (**1**) and trichloropropylsilane (**2**) were calculated by $^1H$ NMR (70% and trace). **1**. $^1H$ NMR ($C_6D_6$, ppm): 0.84 (m, 2H, SiCH₂), 1.41 (m, 2H, SiCH₂CH₂), 2.78 (t, 2H, $^3J_{HH}$ = 6.6 Hz, CH₂Cl).

**Gram-scale synthesis of 1.** A rhodium catalyst **11** (80 mg, 0.060 mmol) was placed in a three-necked flask (3 L) with condenser under a nitrogen atmosphere. Allyl chloride (187 g, 2.45 mol) and trichlorosilane (332 g, 2.45 mol) were added to the flask, and the mixture was stirred at 60 °C for 20 h. After distillation of the

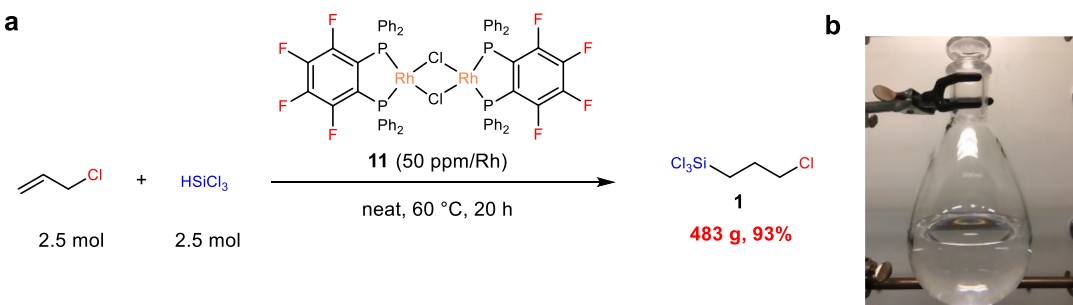

**Fig. 5 Gram-scale synthesis of 1 catalysed by 11. a** Hydrosilylation reaction of allyl chloride with $HSiCl_3$ on a 2.5 mol scale. **b** Trichloro(3-chloropropyl) silane (**1**) after distillation.

resulting solution under the reduced pressure (36 hPa, 80 °C), analytically pure **1** was obtained as a colourless liquid (483 g, 2.28 mol, 93%).

## Data availability

$^1H$, $^{13}C\{^1H\}$, and $^{31}P\{^1H\}$ NMR spectra of the compounds are shown in Supplementary Figs. 2–32. The data that support the findings of this study are available within the paper and its Supplementary Information, and also from the corresponding author upon reasonable request.

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

## Acknowledgements
This work was supported by the "Development of Innovative Catalytic Processes for Organosilicon Functional Materials" project (PL: K. Sato) from the New Energy and Industrial Technology Development Organization (NEDO).

## Author contributions
K.I. performed the experiments and wrote the manuscript. Y. Naganawa discussed the results and wrote the manuscript. Z.A.W. and K. Sakamoto performed some experiments. K.M. planned the initial experiments. K. Sato conceived, designed, and directed the research. Y. Nakajima designed, directed the research, and wrote the manuscript.

## Competing interests
The authors declare no competing interests.
