## [Peer Review File · Communications Chemistry]

Reviewers' comments:

Reviewer #1 (Remarks to the Author):

This research describes hydrosilylation of allylic chloride. The product has been a useful coupling agent for a variety of academic and industrial applications. High efficiency of the reaction was achieved through development of electron poor bidentate Rh catalyst and it showed significantly better reactivity and selectivity vis-à-vis Pt catalysts. Thorough mechanistic investigations shown in this paper revealed a few important aspects of the catalytic processes. Finally, a gram scale synthesis of the product was demonstrated (typo in page 19, must be mol, not mmol).

In a mechanistic standpoint (page 19), the proposed mechanism is interesting and in turn closely related to one from ref 53, in particular, formation of intermediate B, in equilibrium with pi-allyl Rh 12. One curious thing that I found in this interesting catalytic silylation chemistry is that how silyl rhodation takes place in the catalytic cycle. Specifically, this paper proposes that intermediate C undergoes silyl rhodation with the second allylic chloride to form intermediate D. I was wondering what if C just undergoes silyl rhodation within the coordination sphere to form $(\text{dppbzF})\text{RhHCl}(\text{ClCH}_2\text{CHCH}_2\text{SiCl}_3)$. Subsequent reductive elimination provides the product and regenerates intermediate B through the association with allylic chloride (the order of this event can be altered.). Did authors investigate a kinetic study on the reaction, to be able to sort out those two possible pathways.

My second question is about silylative olefin-walking. Based on the proposed mechanism, intermediate D can undergo beta-hydride elimination in two ways, leading to two dehydrogenative silylation products, vinyl chloride and vinyl silane. Did authors observe those byproducts? I think that the second allylic chloride can occupy the coordination site, as proposed, because of the electron-poor rhodium center, and the coordination possibly prevents beta-hydride elimination, which led to preferential reductive elimination. Reinsertion of a resulting dihydridorhodium species to either the olefin in the dehydrogenative silylation products or the starting allylic chloride is indeed possible though.

In SI, ^1H NMR of compound 14 indicates that RhH is dd (-6.17 ppm). Is this supposed to be dt? In the corresponding NMR spectrum provided does not include that peak, possibly that the error occurred during the preparation of the document. Thus, it is not possible to judge whether that is dd or dt. Overall, I feel that authors did an excellent job by performing very detailed mechanistic studies systematically and demonstrating scalability.

Reviewer #2 (Remarks to the Author):

The studies presented in the manuscript COMMSCHEM-20-0369-T concern the assessment of the influence of the chemical structure of ligands in Pt and Rh complexes on their effectiveness in the hydrosilylation process of allyl chloride with trichlorosilane. The authors also attempted to propose the mechanism of the discussed process and to analyze the accompanying side processes.

I find the research subject to be justified from the point of view of industrial and academic applications, however, a few aspects pointed below raise my reservations. Therefore, I believe that the presented manuscript is not suitable for publication in Communications Chemistry in its present form and should be reconsidered after revision.

1. Although the authors discuss the influence of the complex structure on the yield of the obtained

products based on the NMR technique, they did not precisely present the procedure for calculating the yield and selectivity of the studied process. The relevant NMR spectra allowing the verification of the obtained research results are also not presented.

2. In the section entitled "Identification of the catalytically active rhodium complexes." the authors discuss the influence of the structure of rhodium complexes with 1,3-bis (diphenylphosphino) propane and 1,2-bis (diphenylphosphino) benzene as ligands, the highest catalytic activity of which was demonstrated by binuclear complexes with two bidentate ligands $[\text{Rh}(\mu\text{-Cl})(\text{dppp})]_2$ (3) and $[\text{Rh}(\mu\text{-Cl})(\text{dppbz})]_2$ (5). And at the end of the paragraph on page 10, line 3, the authors write: "Based on these results, we postulated a mononuclear Rh (I) species with one bidentate-phosphine ligand as the active species in such hydrosilylation reactions." which is inconsistent with the presented research results.

3. The studies of the catalytic activity and selectivity of the discussed complexes were carried out with their various amounts, at different times, and with different molar ratios of the substrates. From this point of view, the presented TON values are difficult to compare, and, in my opinion, they do not represent a great value. The determination of TON is of great value when comparing the activity of catalysts in the case of their multiple-use when the process is carried out or repeated until their deactivation.

4. In the study of the catalytic activity of the discussed complexes, the authors also omitted the important aspect of the process, which is its time and temperature, which are particularly important concerning the used substrates. The information on how the activity of the complexes changes with temperature and how the conversion of substrates changes over time has not been presented. This information is, in my opinion, crucial for the optimization of an industrial chemical process. Why was the gram-scale synthesis carried out for 35 hours instead of 20 as in the catalytic study?

5. It would also be appropriate to describe the thermal effects accompanying the hydrosilylation process in the presence of the proposed complex.

Reviewer #3 (Remarks to the Author):

This manuscript developed one new rhodium complex to catalyze the hydrosilylation reaction of allyl chloride with HSiCl_3 in high efficiency. Mechanistic studies revealed the importance of the fluorinated dppbz ligand in stabilizing the active rhodium species in the catalytic cycle. The wide use of the trichloro(3-chloropropyl)silane product in industry and the high turnover number and selectivity of this reaction might make this new developed rhodium catalyst a good candidate for industrial use. However, this kind of reaction is known. A similar catalytic system $[\text{Rh}(\text{COD})\text{Cl}]_2/\text{dppp}$ has been reported to catalyze the reaction of HSiCl_3 with allyl chloride in 2019 as a patent (Wang Zhian et al. Jpn. Kokai Tokkyo Koho, 2019085352). Besides, the substrate scope shown in this work is too narrow. Iridium complexes were also found to be efficient in promoting the hydrosilylation reaction of functionalized allyl compounds, but the authors gave few comments on this. Though this work might be interesting for industry, it's not suitable for publishing in Commun. Chem.

Dr. Yumiko NAKAJIMA

Phone: +81-29-861-6707 Fax: +81-29-861-4568
e-mail: yumiko-nakajima@aist.go.jp

February 19, 2021

RE: Manuscript ID: COMMSCHEM-20-0369-T

Title: "Selective hydrosilylation of allyl chloride with trichlorosilane: Investigation into an industrially important issue"

Authors: Koya Inomata, Yuki Naganawa, Zhi An Wang, Kei Sakamoto, Kazuhiro Matsumoto, Kazuhiko Sato, and Yumiko Nakajima

Dear Reviewers,

Thank you very much for giving us various fruitful comments, regarding the above referred manuscript. Based on the comments, the manuscript has been revised by changing all the suggested points accordingly. We hope that the manuscript has been improved satisfactorily to be acceptable for publication in *Communications Chemistry*.

Sincerely yours,
Yumiko Nakajima

Answers to Reviewer 1:

1. *One curious thing that I found in this interesting catalytic silylation chemistry is that how silyl rhodation takes place in the catalytic cycle. Specifically, this paper proposes that intermediate C undergoes silyl rhodation with the second allylic chloride to form intermediate D. I was wondering what if C just undergoes silyl rhodation within the coordination sphere to form (dppbzF)RhHCl(ClCH₂CHCH₂SiCl₃). Subsequent reductive elimination provides the product and regenerates intermediate B through the association with allylic chloride (the order of this event can be altered.). Did authors investigate a kinetic study on the reaction, to be able to sort out those two possible pathways.*

Thank you very much for the beneficial comments. We performed several reactions and finally figured out that the following reaction evidenced the suggested mechanism, which proceeds silyl rhodation without additional coordination of allyl chloride: i.e. the hydrosilylation of allyl chloride with HSiCl₃ in the presence of stoichiometric amount of [Rh(μ -Cl)(dppbz^F)₂] resulted in the selective formation of the hydrosilylated product **1** (84% NMR yield) as well as **2** (8%) (Scheme SS1). When the reaction was performed using the increased amount of allyl chloride to 2 equiv/Rh, **1** was also formed with the similar yield of 89% and selectivity. These results

demonstrated that the silyl rhodation proceeded without the additional coordination of allyl chloride.

Based on the above results, we modified mechanism in the main text (Fig. 4; page 18, line 7-9).

To fully understand the reaction mechanism, we recently started DFT calculations. The intermediate $[\text{Rh}(\text{Cl})(\text{H})(\text{SiCl}_3)(\text{dppbz}^{\text{F}})]$ (**B**) in Fig. 4 was successfully optimized as a reasonably stable form (Figure SS1). On the other hand, it was difficult to get a stable structure after π -coordination of allyl chloride to **B**, possibly due to the steric hindrance against the bulky dppbz^{F} ligand. This result also supports the revised mechanism shown in Fig. 4. The further DFT study is now underway to elucidate the whole reaction mechanism. We hope that we can report the mechanistic details as well as the practical role of dppbz^{F} in the near future.

method: b3lyp

- My second question is about silylative olefin-walking. Based on the proposed mechanism, intermediate **D** can undergo beta-hydride elimination in two ways, leading to two dehydrogenative silylation products, vinyl chloride and vinyl silane. Did authors observe those byproducts? I think that the second allylic chloride can occupy the coordination site, as proposed, because of the electron-poor rhodium center, and the coordination possibly prevents beta-hydride elimination, which led to preferential reductive elimination. Reinsertion of a resulting dihydridorhodium species to either the olefin in the dehydrogenative silylation products or the starting allylic chloride is indeed possible

though.

We did not observe distinct formation of dehydrogenative silylation products in any catalytic reactions shown in Tables 1 and 2. In addition, the above reactions shown in Scheme SS1 demonstrated that the product selectivity was not dependent on the amount of allyl chloride. Thus, we concluded at this moment that the additional coordination of allyl chloride does not bring significant effect on the product selectivity. It was also supported by our preliminary calculation results, showing that additional coordination of allyl chloride to intermediates **B** and **C** in Fig. 4 is difficult due to steric hindrance.

- 3. In SI, ¹H NMR of compound 14 indicates that RhH is dd (-6.17 ppm). Is this supposed to be dt? In the corresponding NMR spectrum provided does not include that peak, possibly that the error occurred during the preparation of the document. Thus, it is not possible to judge whether that is dd or dt.*

The hydride signal of compound **14** at -6.17 ppm is dd (page S15 in SI). The signal shape is now shown in the inset window in the SI, page S46. Based on the referee's comment, we realized that RhH was assignable to the hydride at the basal plane, which exhibits $^2J_{\text{HP}^{\text{trans}}} = 115$ Hz and $^1J_{\text{HRh}} = 15.6$ Hz. Rather small $^2J_{\text{HP}^{\text{cis}}}$ (normally 4-18 Hz) is obscured in this case. For normal $^2J_{\text{HP}^{\text{cis}}}$ values, please see ref 15 in SI. According to this assignment, the structure of **14** was corrected (main text page 17, Fig. 3).

Considering the structure of **14**, which possesses the hydrido ligand on the basal plane, there is a possibility that the hydrosilylation reaction could follow Chalk-Harrod mechanism, where allyl chloride inserts into the Rh-H bond. Through the theoretical mechanistic study mentioned above, this point is also now studied in our team. The related comment is added in the main text, ref 67.

- 4. typo in page 19, must be mol, not mmol*

It was corrected accordingly (page 20, Fig. 5).

Answers to Reviewer 2

- 1. Although the authors discuss the influence of the complex structure on the yield of the obtained products based on the NMR technique, they did not precisely present the procedure for calculating the yield and selectivity of the studied process. The relevant NMR spectra allowing the verification of the obtained research results are also not presented.*

The yield of **1** was determined by the integral intensity ratio of the CH₂ signal at 2.78 ppm towards the signal at 2.15 ppm of mesitylene as an internal standard. This is now described in the SI (page S7, line 6-7) with relevant NMR spectra (page S7, Figure S1).

- 2. In the section entitled "Identification of the catalytically active rhodium complexes." the authors discuss the influence of the structure of rhodium complexes with 1,3-bis (diphenylphosphino) propane and 1,2-bis (diphenylphosphino) benzene as ligands, the*

highest catalytic activity of which was demonstrated by binuclear complexes with two bidentate ligands [Rh(u-Cl)(dppp)]₂ (3) and [Rh(u-Cl)(dppbz)]₂ (5). And at the end of the paragraph on page 10, line 3, the authors write: "Based on these results, we postulated a mononuclear Rh (I) species with one bidentate-phosphine ligand as the active species in such hydrosilylation reactions." which is inconsistent with the presented research results.

Thank you very much for pointing out our confusing phrases. The binuclear complexes behave as a good catalyst precursor to offer the active mononuclear Rh(I) species. To clarify this point, we added the following sentence "Based on these results, we postulated that chloro-bridged dinuclear complex **5**, which could form a reactive mononuclear dppbz^F-Rh (I) species, behave as a good catalyst precursor" in page 10 line 5-7.

3. *The studies of the catalytic activity and selectivity of the discussed complexes were carried out with their various amounts, at different times, and with different molar ratios of the substrates. From this point of view, the presented TON values are difficult to compare, and, in my opinion, they do not represent a great value. The determination of TON is of great value when comparing the activity of catalysts in the case of their multiple-use when the process is carried out or repeated until their deactivation.*

Thank you very much for the valuable comments. We removed all the TON values from the Tables. Now one TON value remains where no increment of the reaction yield was observed after further reaction time (page 13 line 2-3 and ref 51).

4. *In the study of the catalytic activity of the discussed complexes, the authors also omitted the important aspect of the process, which is its time and temperature, which are particularly important concerning the used substrates. The information on how the activity of the complexes changes with temperature and how the conversion of substrates changes over time has not been presented. This information is, in my opinion, crucial for the optimization of an industrial chemical process. Why was the gram-scale synthesis carried out for 35 hours instead of 20 as in the catalytic study?*

The results of the reactions at temperatures and reaction time are now listed in Table 2, entries 10-12. Related with this, some discussion comments were added in the main text (page 12 line 14-17). The reaction time of the gram-scale synthesis was performed for 20 h, affording the same results. This point was also corrected in the main text (page 20 Fig. 5; page 22, line 8).

5. *It would also be appropriate to describe the thermal effects accompanying the hydrosilylation process in the presence of the proposed complex.*

Related to above comments, some comments were now added in page 12 line 14-17. In addition,

we tried to follow reaction kinetics to see the time-course conversion of the hydrosilylation process catalyzed by **11** at different temperatures. However, the catalytic activity of **11** was somehow suppressed whenever we opened the reaction vessel during the reactions in order to get a small portion of the sample for yield evaluation. Such the phenomena also occurred even when the sampling was performed at heated conditions via a septum sealed flask without opening the Schlenk tube. As a result, all the kinetic experiments performed this time were low yielding (ca. 35%). Since the highest catalytic performance was achieved under neat conditions, we could not use in-situ NMR experiments for kinetics. Overall, the kinetic study for the detailed discussion on thermal effects is at this moment not successful.

Answer to Reviewer 3:

- 1. The wide use of the trichloro(3-chloropropyl)silane product in industry and the high turnover number and selectivity of this reaction might make this new developed rhodium catalyst a good candidate for industrial use. However, this kind of reaction is known. A similar catalytic system ($[Rh(COD)Cl]_2/dppp$) has been reported to catalyze the reaction of $HSiCl_3$ with allyl chloride in 2019 as a patent (Wang Zhian et al. Jpn. Kokai Tokkyo Koho, 2019085352). Besides, the substrate scope shown in this work is too narrow. Iridium complexes were also found to be efficient in promoting the hydrosilylation reaction of functionalized allyl compounds, but the authors gave few comments on this. Though this work might be interesting for industry, it's not suitable for publishing in Commun. Chem.*

Thank you very much for the valuable comments. The patent work that suggested by the reviewer is indeed performed by our group. This is important pre-work of this study and now cited as ref 40 (page 5). As pointed out by the reviewer, Ir complexes are known as good catalyst precursors for hydrosilylation of allylic compounds. Some of the works have been also reported by our group. These are now cited as ref 24, 34, 42-44 (page 6). Surprisingly, none of the reported Ir catalysts were applicable for the selective hydrosilylation of allyl chloride with $HSiCl_3$. Then, we realized the importance to focus on the development of novel catalyst system, which is especially oriented to the reaction of allyl chloride with $HSiCl_3$. This point is now explained in the main text (page 6, line 12-15).

Throughout our continuous study of hydrosilylation reactions, we thus far focused on the synthesis of various industrially important organosilicon compounds. As a result, we realized how difficult to overcome the limitation of conventional Pt catalyst systems and how important to develop new catalyst systems, which achieve hydrosilylation of only limited substrates but enable us to efficiently synthesize target compounds, for the purpose to contribute the silicon industry. We believe this piece of work will bring one answer to this industrially important issue from the academic view point and could open up new research avenues in this research area.

REVIEWERS' COMMENTS:

Reviewer #1 (Remarks to the Author):

I feel that authors tried to address this reviewer's concerns and comments and did well during this pandemic. The revised mechanistic proposal seems to be more appropriate to justify the chemistry, with help of the additional experiments and quick computation. No other comments.

Reviewer #2 (Remarks to the Author):

I am satisfied with the changes made to the manuscript and can now recommend its publication in Communications Chemistry in its present form.